# Application of Composts’ Biochar as Potential Sorbent to Reduce VOCs Emission during Kitchen Waste Storage

**DOI:** 10.3390/ma16196413

**Published:** 2023-09-26

**Authors:** Joanna Rosik, Jacek Łyczko, Łukasz Marzec, Sylwia Stegenta-Dąbrowska

**Affiliations:** 1Department of Applied Bioeconomy, Wrocław University of Environmental and Life Sciences, Chełmońskiego Str. 37a, 51-630 Wroclaw, Poland; 118318@student.upwr.edu.pl (J.R.); 116159@student.upwr.edu.pl (Ł.M.); 2Faculty of Biotechnology and Food Science, Wroclaw University of Environmental and Life Sciences, 50-375 Wroclaw, Poland; jacek.lyczko@upwr.edu.pl

**Keywords:** VOCs, kitchen waste, waste storage, compost biochar

## Abstract

It is expected that due to the new European Union regulation focus on waste management, managing kitchen waste will become more important in the future, especially in households. Therefore, it is crucial to develop user-friendly and odour-free containers to store kitchen waste. The study aimed to test the effectiveness of composts’ biochar in reducing noxious odours and volatile organic compounds (VOCs) released during kitchen waste storage. Various amounts of compost biochar (0%, 1%, 5%, and 10%) were added to food waste samples and incubated for seven days at 20 °C. The released VOCs were analysed on days 1, 3, and 7 of the storage simulation process. The results indicated that adding 5–10% of composts’ biochar to kitchen waste significantly reduced the emissions in 70% of the detected VOCs compounds. Furthermore, composts’ biochar can be used to eliminate potential odour components and specific dangerous VOCs such as ethylbenzene, o-xylene, acetic acid, and naphthalene. A new composts’ biochar with a unique composition was particularly effective in reducing VOCs and could be an excellent solution for eliminating odours in kitchen waste containers.

## 1. Introduction

Kitchen waste is a group of biodegradable waste that includes peelings, fruit, and vegetable leftovers, out-of-date food products, as well as coffee grounds and tea leaves. The main problems of waste management in terms of kitchen waste management are the emission of odorous volatile organic compounds (VOCs), greenhouse gases such as benzene, chloroethylene, or styrene [1], and the presence of pathogenic microorganisms such us *Bacillus*, *Pseudomonas*, or *Flavobacterium* [2]. The rapid degradation of kitchen waste during home collection is caused by micro-organisms through metabolic processes that contribute to the decomposition of organic matter and the emission of odours. Short-term exposure to proximity to a VOC emitter causes dizziness, headaches, nausea, and irritation of the skin and nose. In addition, prolonged exposure to an emitter can increase the risk of cancer [3]. Substances emitted from the compost pile also contribute to environmental pollution and negatively affect natural ecosystems [4]. Due to the harmfulness of VOCs on the environment and human health (in extreme cases, even life) [5] new solutions are being sought to help neutralise them. Current technologies that may contribute to the prevention of VOCs emissions in the future include membrane separation [6], thermal catalysis [7], non-thermal plasma [8], and adsorption [9]. Currently, some studies test the effectiveness of combining methods such as plasma catalysis [10]. These are developing technologies, nevertheless, they can become very useful tools for VOC removal because of their high efficiency. In some cases, the elimination of VOCs by the abovementioned methods reached up to 90%. Unfortunately, because of the lack of universality of these methods, the costs associated with their possibility of use are very high.

Adsorption is one of the most common methods for removing VOCs because of its ease of use and low cost. The most popular adsorbent for harmful volatile substances is activated carbon, but more and more studies are emerging that also report the high effectiveness of biochar [11,12,13]. 

Duan et al. [14] investigated the effectiveness of a bamboo biochar addition to chicken manure compost on the degradation of volatile fatty acids. The study made it possible to observe a high correlation between the adsorption of harmful substances and the physicochemical properties of biochar, such as porosity. The addition of biochar to the compost mass caused a reduction in the fatty acid content by converting acid fats into humic acids. Increasing the HA/FA proportion (which was 4.92) resulted in an accelerated humification process, faster stabilisation of the compost, and partial neutralisation of odours. Sánchez-Monedero et al. [15] used oak biochar to test its effectiveness in the context of reducing VOCs emissions in chicken manure and straw compost. The results showed that the best effect of a 10% biochar addition was observed in the thermophilic phase of the process. The study proved that biochar prevents the formation of anaerobic sites in the compost mass, which can be a source of VOC emissions.

The environmental conditions inside the pile may not always be favourable for the sorption of VOCs. Competition of water vapour and organic molecules for adsorption sites in biochar can lead to a reduction in the sorption capacity of the material. Hwang et al. [16] conducted a study on the potential of biochar to remove VOCs from swine feces. Biochar from poultry litter, pig manure, oak, and coconut shells (each substrate was pyrolysed at 350 °C and 500 °C) and activated carbon produced from coconut shells were used in the study. The results presented in the research showed that oak biochar pyrolysed at 350 °C revealed the highest sorption capacity, but less than the activated carbon from the coconut shell. This suggests that the pyrolysis temperature has a significant effect on the sorption capacity of biochar. Awasthi et al. [17] studied the effect of a biochar addition on volatile fatty compounds and odour release during the composting of sewage sludge. The study shows that biochar can optimise the bulk density of the pile and its porosity, which has a positive effect on the abundance of microorganisms. Increasing the number of microorganisms such as lactic acid bacteria resulted in an acceleration of the process and its intensification, which in turn enabled the faster stabilisation of the process. As organic matter decomposed, the number of microorganisms in the pile and the amount of volatile fatty acids (VFAs) emitted also decreased. The intensification of the process, as a result of increased microbial activity, especially in the thermophilic phase, led to a reduction in the abundance of long-chain compounds such as valerian acid. The authors also observed a positive correlation between temperature and VOCs emissions and a negative correlation between temperature and VFAs emissions. Czekała et al. [18] investigated the effect of adding biochar to poultry manure compost on temperature and carbon dioxide emissions during the process. It was observed that the addition of biochar increased the process temperature and shortened the thermophilic phase. Additionally, increasing the process temperature was responsible for increasing the proportion of carbon dioxide emitted.

In each of the studies mentioned, a positive relationship was observed between the addition of biochar and compost parameters, such as pH, temperature, aeration, and bacterial count, which may not only help to control the composting process, but also to effectively neutralise VOCs. The biochar added to the compost caused those in the piles to not form free of oxygen, which caused the onset of the adverse process of anaerobic digestion. Physicochemical properties of biochar such as porosity caused the pile to be optimally aerated, which resulted in a balance of transformations in the compost. Due to this, parameters such as pH, temperature, and number of bacteria were stabilised, which had a positive effect on the maturation of compost. The popularity of biochar has also influenced the emergence of research that leads to improved physicochemical properties of biochar for the more efficient adsorption of VOCs. Zhang et al. [19] noted that the ball milling of biochar with ammonia hydroxide or hydrogen peroxide improves the adsorption of phenyl VOCs. The study involving the modification of biochar improved, among other things, the volume of micropores responsible for the adsorption process of biochar and made the adsorbent more homogeneous. The larger volume of micropores promoted the easier diffusion into the pores and transport of the microparticles of some organic compounds, such as benzene, to the adsorption sites.

Due to the beneficial environmental impact determined by the sorption capacity and the preparation method, biochar can also be considered as part of a circular economy model [20,21]. Thermal conversion via the pyrolysis process helps to reduce the amount of waste [22]. Additionally, a by-product of the pyrolysis process is biochar, whose properties can contribute to the removal of harmful odour compounds from the air, but adsorption is not its only application. The presence of pores on the surface of biochar makes the material suitable for use in the manufacturing of construction materials as a filler and cement replacer in cementitious materials [23]. The possibility of using biochar in agriculture as an additive to improve soil properties has also been observed, which can become an alternative to conventional fertilisers [24]. Biochar-based fertilisers are characterised by a slow release of micro and macro elements into the soil. Slower nutrients release prevents the loss of valuable minerals [25]. Additionally, biochar, due to its adsorption properties, can retain heavy metals present in the soil on its surface [26]. The specific properties of biochar and the possibilities of its use make it a part of sustainable agriculture [27]. Recent studies also emphasise the important role of the non-carbonated fraction as another VOC sorption mechanism in biochar, which could be derived from materials with low pyrolysis or a low organic content used for biochar production [12]. Recent studies have shown great potential to convert waste into carbon-based compounds in the near future [28]. Although organic waste has been widely researched for its potential in biochar production, especially in high-temperature pyrolysis [29], the use of low organic components such as compost, which are rich in nutrients and inorganic substances—a source of non-carbonated fractions—is an area that has not been explored in terms of VOC adoption. For this reason, compost that originates from home composting has been used to produce biochar in low-temperature slow pyrolysis (composts’ biochar).

This study aims to test the composts’ biochar properties and the ability of composts’ biochar to reduce the VOCs compounds and noxious odour released during kitchen waste storage. It is expected that due to the new European Union (EU) regulation focus on waste management, managing kitchen waste will become more important in the future, especially in households. Therefore, it is essential to develop easy-to-use and odour-free containers for kitchen waste. Various measurements must be followed to manage the respective malodorous emissions. The purpose of the study is also to compare the physicochemical properties of biochar and observe the relationship between the amount of inorganic matter that builds the structure of compost biochar and the adsorption capacity of VOCs.

## 2. Materials and Methods

### 2.1. Materials

In the study, kitchen waste and composts’ biochar were used. A mixture of kitchen waste was prepared with the following proportions, according to Cesaro et al. [30]:vegetable and fruit waste, 65% by weight;pasta, rice, and bread waste, 20% by weight;meat and dairy waste, 15% by weight.

To obtain homogeneity under laboratory conditions and repetitive data, kitchen waste was ground in an electric mill (Bosch SmartPower 1500W, MFW2514W, BSH, Warsaw, Poland) with a mesh diameter of 3.5 mm. Composts’ biochar was prepared using a laboratory muffle furnace (SNOL, model 8.1/1100, Utena, Lithuania) in retention time—1 h, temperature 350 °C, heating rate of 50 °C∙min^−1^. Material used for composts’ biochar production was compost originating from a home composter (Ekobat, ecosmart 1000; Wroclaw, Poland). CO_2_ was supplied into the chamber during the entire pyrolysis process to maintain an inert atmosphere. After carbonisation, the furnace was turned off and left to cool. For this study, the study biochar was ground in a mill using a 1 mm sieve ground by a 1 mm-knife knife mill through a 1 mm screen (Testchem, LMN-100, Pszów, Poland). The material was stored in plastic containers at room temperature. 

### 2.2. Experimental Procedure

Experiments were carried out in small glass reactors, 100 mL volume, with approximately 50 g of kitchen waste in each. The composts’ biochar was added to kitchen waste samples 0%, 1%, 5%, and 10% on a wet basis (Figure 1). The reactors with material were incubated for 7 days at a temperature of 20 °C in a laboratory incubator (POL-EKO-APARATURA, model ST 3 COMF, Wodzisław Śląski, Poland) to simulate the storage of kitchen waste in household containers. Seven days was an estimated time to collect kitchen waste under domestic conditions. Basic analyses were provided for each component and material. For each waste component (kitchen waste and composts’ biochar) and in each reactor before and after the process basic analyses were provided, including pH, electrical conductivity (EC), moisture content (MC), and volatile solids (VS) content. Kitchen waste and compost biochar were additionally subjected to FTIR and VOC analysis.

### 2.3. Methods

#### 2.3.1. Materials Analysis

Moisture content (MC) was measured at a temperature of 105 °C using a laboratory dryer (WAMED, model KBC-65W, Warsaw, Poland). To determine the loss on ignition, (LOI) organic matter content at 550 °C, 4 h held, a laboratory muffle furnace (SNOL, model 8.1/1100, located in Utena, Lithuania) was used. Electrical conductivity (EC) and pH were determined using a pH-meter (Elmetron, CPC-411, Zabrze, Poland) in a 1:10 water solution [1]. The cation exchange capacity (CEC), which was determined as the sum of base cations, was measured on a microwave plasma-atomic emission spectrometer (MP-AES 4200, Agilent Technologies, Santa Clara, CA, USA) at pH 7.0 after extraction with 1 M ammonium acetate.

For augmented total reflection-Fourier transform infrared (ATR)-FTIR measurements, a Nicolet iN10 integrated infrared microscope with Nicolet iZ10 external FT-IR module (Thermo Fischer Scientific, Waltham, MA, USA) used a deuterated-triglycine sulfate (DTGS) detector and a diamond ATR module. For each spectrum, 32 scans were averaged in the mid-IR range of 400–4000 cm^−1^.

#### 2.3.2. Sample Preparation for VOC Analysis

VOC analysis was performed on days 1, 3, and 7 of the storage simulation process. For the analysis, ~0.5 g mixture of kitchen waste and biochars were placed in 20 mL dark glass vials, and sealed with a PTFE septum and with extruded aluminium caps. A piece of 2 × 2 cm filter paper with 3 μg of 1 mg‧mL^−1^ caryophyllene was placed in each vial which facilitated the quantification of VOCs. The caryophyllene was placed on the paper filter, and not directly on the material, due to the possibility of absorbing it by biochar.

#### 2.3.3. VOC Analysis—GC–MS, SPME Arrow Extraction

Gas chromatography–mass spectrometry (GC–MS) combined with solid phase microextraction in the gas phase (SPME Arrow, Arrow fiber, Shimadzu, Kyoto, Japan) was used for the separation, identification, and quantification of volatile organic compounds (VOC). VOCs were performed with Shimadzu GCMS QP 2020 Plus (Shimadzu, Kyoto, Japan) equipped with a ZB-5Msi capillary column (30 m × 0.25 mm × 0.25 µm; Phenomenex Ltd., Torrance, CA, USA).

The operational conditions of the GC were as follows: injection port 50 °C; temperature program started at 50 °C and held for 2 min, then at the of rate 3 °C·min^−1^ to 180 °C, then at the rate of 20 °C·min^−1^ to 270 °C, held for 5 min, 10 s; helium as carrier gas with flow 1 mL·min^−1^; split 100 (SPME Arrow analysis).

The extraction of VOCs was performed with 1.10 mm DVB/C-WR/PDMS SPME Arrow fibre (Supelco, Bellefonte, PA, USA). Extraction was carried out in 20 mL headspace vials for 30 min at 45 °C. The extraction proceeded with incubation for 10 min at the same temperature. The analytes were desorbed under the conditions of the GC injection port for 3 min. MS operational conditions were as follows: interface temperature 250 °C; ion source temperature 250 °C; scan 40–400 *m*/*z*.

### 2.4. Statistical Analysis

For statistical analysis, 13.3 Statistica software (TIBCO Software Inc., Palo Alto, CA, USA) was used. For VOCs contribution statistical differences, one-way ANOVA was applied according to Tukey’s test at a significance level *p* < 0.05, including previous verification of normality and homogeneous variance using the Levene test. For all relevant cases, standard deviation (SD) was applied.

## 3. Results and Discussion

### 3.1. Properties of Compost, Biochar, and Kitchen Waste

The amounts of individual elements and heavy metals contained in the compost and in the biochar made from the compost were compared (Figure 2). For all elements, a slight increase in their content was observed in the biochar compared to the compost. The reason for the increase in the proportion of elements is the thermal transformation of the material. Subjecting the material to pyrolysis at 350 °C for 1 h concentrated the substances present in the material, thus increasing the proportion of elements studied.

The amount of N in the compost was 0.73% d.m.^−1^, while after the pyrolysis process, it increased to 1.19% d.m.^−1^. The total content of P in the compost reached 0.47% d.m.^−1^ while the biochar was characterised by a P content of 0.68% d.m.^−1^. In the case of K, an increase in the share of this element was also evident. From a value of 1.7% d.m.^−1^, K increased to 2.1% d.m.^−1^. In a study that Phares et al. [31] conducted, the opposite trend was observed for N, P, and K—only 30–40% of N, P, and K remained in biochar after pyrolysis. A potential reason for the differences in results could be the different material from which the biochar was made, as well as the temperature at which the pyrolysis was conducted and its length. The sample was pyrolysed at 550 °C for 4 h. On the other hand, a study conducted by Konak et al. [32], which tested the effect of temperature on the physicochemical properties of the substrates, showed that biochar made at 300 °C had the highest accumulation of nutrients. For each temperature, an increase in minerals relevant to soil application was observed. The greatest difference between compost and biochar was observed for C, as it increased from a value of 9.21% d.m.^−1^ to 18.03% d.m.^−1^. On the other hand, Na before the pyrolysis process in compost was 4.45 mg·kg d.m.^−1^, while after the thermal treatment of the sample, it reached a value of 6.41% d.m.^−1^. Mn increased slightly after the sample was subjected to the pyrolysis process, while its amount before and after the process was insignificant. A Mn value of 0.26% d.m.^−1^ was observed in the compost, while in the biochar, this value increased to 0.34% d.m.^−1^. As the amount of C increased in the biochar tested, the C/N parameter also increased. In the compost, its value was 12.69 (indicating the complete humification of the material), after which it increased to 15.15.

Similarly, for biologically useful elements, an increase in heavy metals such as As, Cu, Al, Ni, B, Cr, and Pb was observed. Chemical analysis showed that the heavy metals with the highest amounts in the sample were Cu and Pb. The high values of these heavy metals indicate the use of pesticides during the cultivation of composted vegetables and fruits. Cu in the compost was 50.36 mg·kg d.m.^−1^, while in the biochar it was 61.48 mg·kg d.m.^−1^. The Pb for the compost reached a value of 85.95 mg·kg d.m.^−1^, while in the biochar it was 99.4 mg·kg d.m.^−1^. The elements with the smallest amounts were As and Al. For compost, they obtained values of, respectively, 3.64 mg·kg d.m.^−1^ and 5.05 mg·kg d.m.^−1^, and for biochar, 4.06 mg·kg d.m.^−1^ and 7.28 mg·kg d.m.^−1^. Medyńska−Juraszek et al. [33] studied the effects of compost and biochar on changes in the mobility of heavy metal absorption by green leafy vegetables. Green waste from the city was used as the material used in the composting process, while the material for biochar was wheat straw, and the material was subjected to a pyrolysis process at 550 °C. In the case of compost, higher amounts of heavy metals were observed than in the case of biochar, which has to do with where the material was obtained. Green waste deposited in urban areas is characterised by a high accumulation of toxic and undesirable elements in plant aids. Heavy metals with the highest concentration in the material were Cu, Pb, and Cr, whose values were 45.3 mg·kg d.m.^−1^, 24.3 mg·kg d.m.^−1^, and 19.0 mg·kg d.m.^−1^, respectively. Addai et al. [34] studied the effects of biochar, compost, and/or NPK fertiliser on the uptake of potentially toxic elements and promoting yields of cultivated lettuce in an abandoned gold mine. The study used poultry manure compost and rice husk biochar made at 500 °C for 3 h. In the study, low values of individual heavy metals were observed in both the compost and biochar. Furthermore, the heavy metals contained in the biochar had a lower proportion than those in the composted material.

The physicochemical parameters of the kitchen waste were examined on days 1 and 7 of the process. On the first day of the process, the kitchen waste was characterised by a moisture content of 80.05%, an organic dry matter content of about 95.53%, a pH of 4.92, and a conductivity of 2.05 mS‧cm^−1^. After 7 days, the material was again analysed and the basic parameters were determined. A slight decrease in the moisture content to 79.95% and an equally slight increase in the organic dry matter content to 95.78% were observed. After 7 days, the acidity of the material increased, as evidenced by a pH of 3.87. The conductivity also decreased and was 1.91 mS‧cm^−1^. What is more important is that the neutral pH of biochar could increase the pH level of kitchen waste, but could also affect the good absorption of P, which is the most effective for biochar with a pH near 7, and the small-dose application [29]. The physicochemical properties of the compost before the torrefaction process and the finished product, which was biochar, were also analysed. The cation exchange capacity (CEC) was examined, whose value in the compost was 143.28 cmol(+)‧kg^−1^. Biochar was characterised by a decrease in the value of CEC compared to the substrate subjected to torrefaction. The CEC for biochar was 123.92 cmol(+)‧kg^−1^. The largest differences were observed for N-NO_3_. The proportion of ammonium ions for compost was 1322.03 mg‧kg^−1^ d.m., while for biochar it was 16.03 mg‧kg^−1^ d.m. N-NH_4_ for compost reached 125.85 mg‧kg^−1^ d.m., while for biochar it was 55.99 mg‧kg^−1^ d.m. Another parameter tested was the LOI (loss on ignition), which was 34% d.m. in the compost, while a lower value of 19% d.m. was recorded in the biochar. The pH was also tested, which reached similar values for both materials. The hydrogen ion ratio in the compost reached a value of 7.09, while a slightly lower value of 6.92 was observed in the biochar. The water content in the compost was 40.20%, while in the torrefied material, the moisture content was only 1.71%. The electrolytic conductivity was examined only in compost and was 2.88 mS‧cm^−1^.

Composts’ biochar was characterised by the composition and emissions of VOCs—51 substances have been recognised (Appendix A). The substance has been identified as a potential hazard and described for its chemical characteristics, such as the use of odours used in agricultural chemicals (Appendix A). The emission of a majority of components of VOCs was very low <50 µg∙kg^−1^ d.m.; however, 23 identified substances are responsible for some hazards—they cause health or/and environmental problems. This indicated that emissions of these substances provide information about special treatment during the storage of this biochar because they could be dangerous to health (e.g., Nonane; Octane, 3-methyl-; Decane, 3-methyl-) when they are inhaled or swallowed. Some suggestions to limit this threat could be the use of palletisation, which can reduce even 80% of emissions [35]. However, this should be studied in future studies to recognise the effect of biochar compost palletisation on the sorption of compounds from VOCs.

It should be noted that biochar compost has much fewer identified substances of the dangerous substances than other known biochars, e.g., municipal solid waste (>80 compounds).

On the other hand, some substances are recognised as important in plant or micro-organism metabolism (octane, 2,3,6,7-tetramethyl-; decane, 2,3,5,8-tetramethyl-; decane, 3-ethyl-3-methyl-; tetradecane <n->), which could be a benefit when used as an addition to soil or waste in the composting process. The greatest risk associated with the use of this type of biochar is its storage and way of application. The substance with the biggest emission such as bicyclo [2.1.1]hex-2-ene, 2-ethenyl-, 444.7 µg∙kg^−1^ d.m.; 1,3-cyclopentadiene, 5-(1-methylethylidene)-, 358.9 µg∙kg^−1^ d.m., and azulene 208.1 µg∙kg^−1^ d.m., are not dangerous, or dangerous only with direct contact. This study confirms that these dangerous substances were not observed when biochar is applicable to kitchen waste storage, which could be the result of the bacterial metabolism (other substances were released) or a level of emission that is below the GS-MS and below the detection threshold (Appendix A).

#### Infrared Spectroscopy Analysis of Compost and Biochar

Infrared spectroscopy was used to identify the functional groups present in the sample of compost and biochar produced from compost. The surface functional groups of compost and biochar are displayed in Figure 3. As can be observed, the FTIRs of the organic materials were different from others in terms of intensity and specific functionality.

In the compost, the predominance of the peaks or bands at 3350, 1600–1650, and 1110–1050 cm^−1^ was observed, indicating that the composts were rich in aromatic, phenolic, aliphatic, and polysaccharide structures. The samples’ spectra displayed the typical bands for compost characterisation, as previously shown by Mujtaba et al. [36] and Silva et al. [37]. The band at 3320 cm^−1^ was associated with HC bonds (Figure 3A). Two characteristic bands, asymmetric (CH_2_) 2935 and symmetric (CH_3_) 2859 cm^−1^, were attributed to the asymmetric functional group and the aliphatic functional group. The most remarkable change in FTIR with compost torrefaction is a decrease in the C–H and O–H bands in the region of 2800–3600 cm^−1^ [36,37]. These bands were barely visible in the biochar spectra (Figure 3B).

The aromatic carbon (C=C) vibration and carboxyl (C=O) stretching were found at 1636 cm^−1^ in compost, which was visible also in fruits and vegetables waste [36]; this confirms that compost was mainly produced from kitchen waste. This peak decreased in biochar (Figure 3). The C-H bending of the carbonyl functional group peak was detected in compost at 1417 cm^−1^, while a greater intensity of C-H binding was observed in biochar. The representative transmittance of the aromatic CO or phenolic OH stretching vibration detected at around 1317 cm^−1^ was observed in biochar, but it was not noticeable in compost. The band around 1020 cm^−1^ could be attributed to the combination of the C-O stretching of polysaccharides, as well as the Si-O-Si bonds of silica and the group Si-O-C, according to Rueda, both in compost and biochar [38]. The high transmittance of aromatic C-H stretching associated at 870 cm^−1^ was observed in biochar, whereas it was barely visible in compost. Furthermore, a certain decrease and changes in the shape of the carbohydrate bands in the region 950–1200 cm^−1^ were observed in biochar. Some changes in the composition of organic matter can be hidden by the overlap with the mineral phases (mainly calcium carbonate and silicates).

Rueda et al. [38] suggest that the thermal treatments at different temperatures lead to an evident variation in the organic matter structure of the organic amendments, especially in the case of immature compost.

### 3.2. Emissions of VOC during the Incubation of Kitchen Waste at 20 °C

The results of individual compounds emitted from kitchen waste obtained from GC–MS analysis were grouped according to their basic chemical groups (Figure 4). A total of 87 compounds were identified, including 17 alcohols and phenols, 7 ketones and aldehydes, 29 o-compounds (esters, acids, furans), 4 sulphates, 12 terpenes and terpenoids, 9 aromatic compounds, and 9 unidentified compounds. Some of these families (alcohols, ketones, aliphatic hydrocarbons, nitride molecules, and sulphide molecules) are extensively degraded during the biological process and detected only in the first stages of composting [39].

The O-compounds were the dominant group of compounds highlighted in the study, especially for the control sample, reaching 32.0–77.7% of the total percentage of identified chemical compounds. Volatile organic compounds (VOCs) may arise from the decomposition of kitchen waste, as well as from reactions between alcohols and carboxylic acids [40]. The addition of biochar resulted in a decrease in the percentage of these compounds, especially on the seventh day of testing in each of the samples, while the greatest changes were observed at doses of 5% and 10%. Sánchez-García et al. [41] also observed that the use of biochar reduced the generation of VOC and particularly oxygenated volatile compounds, typically generated during deficient aerobic degradation. The addition of biochar improved the physical properties of the material, preventing the formation of clumps larger than can be inductive to anaerobic conditions in the pile. This effect may explain the effectiveness of biochar in reducing O-compounds.

As the proportion of O-compounds decreased, an increase was observed in the proportion of alcohols and phenols, which were the second most abundant chemical group. Higher doses of biochar were observed to increase alcohol emissions, which were 59.8% on the seventh day for the 5% biochar addition and 59.5% in the sample with a 10% biochar addition. During the early stages of organic decomposition, alcohol is typically produced through anaerobic or poorly aerobic processes [42,43], especially when oxygen could be a limiting factor due to the high microbial activity [44]. The study was conducted under oxygen-limited conditions, as could be observed during kitchen waste collections, so, in particular, the lower layers of the material in the reactors could undergo methane fermentation, resulting in the generation of alcoholic compounds, VOCs from decomposing kitchen waste, along with the reactions that occur between alcohols and carboxylic acids. The observation of semi-aerobic conditions is also evidenced by the presence of valeric acid, typical of the methane fermentation process (Appendix A) [45]. Biochar creates an optimal environment for microorganisms, which accelerates colony growth even in a fermentation process [46]. The phenols present in the study may have come from meat waste, as they are formed by the metabolism of amino acids [47]. Scaglia et al. [48] confirmed that the major part of alcohols is emitted during the first biostabilisation stage when oxygen could be a limiting factor due to the high microbial activity [44]. In fact, in the present study, alcohols were strongly emitted with the addition of biochar, which can be an effect of the stimulation of the activities of specific microbes that can excrete functional enzymes and affect the microbial activity through altering physicochemical properties, such as increasing pH and C/N and facilitating the degradation of kitchen waste.

The third most abundant group were aromatic compounds. A decrease in their percentage over time was observed in the control sample, while the addition of biochar caused the inhibition of this decrease. The aromatic compounds present among the identified chemical groups were most likely derived from vegetables and fruits, whose share in kitchen waste was significant. These compounds are produced from the microbial degradation of organic plant matter or bulking agents during composting [49].

A large reduction was observed for ketones and aldehydes, as well as terpenes and terpenoids, which have been described as the main compounds and are responsible for odorous pollution. It is clearly visible that the addition of biochar has a positive effect on the amount of terpenes, terpenoids, and unpleasant odours during the storage of the kitchen waste. The presence of ketones and aldehydes together with sulphides indicates the incomplete aerobic decomposition of organic matter [47]; terpenes are more related to the presence of vegetables in the kitchen and they are present in peels, fleshes, oils, and juices of citrus fruits [39].

Conditions of incomplete aerobic decomposition are particularly conducive to the accumulation of sulphides, which were most likely present, in particular, in the lower parts of the material in the reactors due to the lack of or low availability of oxygen. For the group of terpenes and terpenoids, the compound with the highest proportion was limonene, which is usually one of the main terpenes detected in kitchen waste [50,51].

It is crucial for users of kitchen waste containers to have easy-to-use and odour-free containers. Therefore, pleasant and unpleasant odours were observed and recorded (Figure 5 and Figure 6; Appendix A). The study made it possible to analyse the emissions of VOC compounds with the highest mass share produced over 7 days for the control sample and the addition of individual biochar doses. A total of 58 compounds were identified in kitchen waste air without added biochar, while 72 compounds were identified in kitchen waste air with added biochar (22 compounds were responsible for pleasant odours, while 10 for unpleasant). Valeric acid, octanol <n->, octanoic acid, and phthalate <diethyl-> (Figure 5 and Figure 6) represented the most typical unpleasant odours during kitchen waste storage [52]. Butyric acid, hexanoic acid, and camphor were observed only in untreated waste, while caproic acid, cresol<meta->, and caprylonitrile (only with biochar addition) were also noted (Appendix A). The results of the current study are in agreement with the emissions measured before in composting plants where alkanes (Decane, Undecane, Tetradecane) (ethanol), esters (Acetic acid, methyl ester; Hexanoic acid, methyl ester; Octanoic acid, methyl ester), alcohols (1-Octanol), terpenes (Alpha-pinene; limonene; Delta.3-carene), and furans (Furan, 2-pentyl) were presented [53].

Valeric acid had no statistically significant effect from biochar addition, but its emission share increased over time, especially with the 5% and 10% biochar doses (Figure 6a). Octanol <n-> (Figure 5b) emissions in the control sample decreased over time. Biochar doses levelled the mass share of octanol on the first day, which was not observed in the untreated sample (Figure 6b). Biochar doses were not a significant factor in reducing octanoic acid emissions (Figure 5c). Phthalate <diethyl-> was significantly reduced, especially with the addition of 5% biochar, while the 1% biochar dose increased its emission values (Figure 5c,d). All doses showed a statistically significant reduction in octanoic acid and phthalate <diethyl-> emissions compared to the control sample. It was observed that the largest unpleasant odours had lower emissions at the end of storage than on the first day (Figure 6a–d). Butyric acid, hexanoic acid, and camphor were only observed in untreated samples and were eliminated with biochar addition. Caproic acid increased during storage, but larger biochar doses limited its emission to a minimum (46.1 μg∙kg^−1^ d.m. after 7 days of storage with 10% biochar addition). Other unpleasant odours (cresol <meta-> and caprylonitrile) were released in small doses of no more than 15 μg∙kg^−1^ d.m. (Appendix A). However, the perception of malodorous substances should not be seen as a result of a single component, but as a synergy or antagonism between compounds [53].

The proportion of pleasant VOCs was also changeable during the storage of kitchen waste. By analysing the emissions of hexanol <n-> (odour of freshly mown grass, Figure 5e) emissions in kitchen waste, it was observed that low doses of biochar were particularly reducing the emissions percentage. The addition of biochar statistically decreased the level of this component of VOCs (Figure 5e). A dose of 1% allowed for an almost complete reduction of hexanol <n-> emissions in the test sample on each test day. Another compound analysed was acetic acid, hexyl ester (mild sweet odour, Figure 5f), whose emissions increased especially on the third day of the study (Figure 6f). In the case of this compound, it was not observed that the biochar dose had a particular effect on the reduction of emission, while the mass share of this compound decreased during the process (Figure 5f). Limonene is not considered a bad odorant; however, it may play an important role in the perception of waste odour in combination with certain microbial volatiles [53]. These emissions in the sample with the biochar addition also showed a decreasing trend (Figure 5g). The emission reduction was best influenced by the 1% and 5% doses. The 10% dose initially showed a significant reduction in limonene emissions, while on the seventh day, the mass share of the compound increased (Appendix A). An inverse relationship was observed for the emissions of benzene-ethanol (Figure 5h), as the emission reductions occurred only in the sample where biochar was applied at a dose of 1%. The other biochar led to a marked increase in emissions of the chemical compound; however, no statistical differences were observed (Figure 5h). During the storage of kitchen waste, the level of the content of limonene and hexanol <n-> were decreased while the benzene-ethanol and acetic acid, hexyl ester increased statistically. At the same time, other pleasant odours were released with the addition of biochar, such as benzene <ethyl->, isoamyl acetate, o-xylene, heptan-2-ol, pseudocumene, caprylic acid, naphthalene, and octanoic acid, ethyl ester (Appendix A), and only five other VOCs were missed when biochar was added: octanal <n->, nonanal <n->; octanoic acid; decanal <n->; and dodecanol. As biochar was successfully used for the elimination of potential odour components, it should be noted that it could also be dedicated to the sorption of specific dangerous VOCs such as ethylbenzene, o-xylene, acetic acid, and naphthalene, as observed in other studies with biochar [16,52]. All of these statements proved that the addition of composts’ biochar has a positive effect in reducing unpleasant odours during the long storage of kitchen waste.

On the other hand, excess exposure to some VOCs may cause mild or serious health problems, as shown in Appendix A. Eleven VOCs compounds were categorised as potential agricultural chemicals, which could influence the final results of biochar addition. Chemical molecules in these chemicals could distract the metabolism of microorganisms in the effective elimination of unpleasant odours. Some of those chemicals are natural components of plants and finally kitchen waste (e.g., limonene, 1-octen-3-ol), but others (e.g., naphthalene, guaiacol), could be an effect of used pesticides in agriculture.

Authors usually emphasise the role of the high sorption capacity of biochar in a mechanism for VOC removal. However, biochar can quickly become saturated when a wide range of VOCs and concentrations are generated [16], limiting its contribution to VOC removal. The environmental conditions, characterised by high moisture, temperature, and organic matter content, may not be optimal for the sorption of VOC on biochar surface, limiting its sorption capacity. Especially high temperatures, such as those typically registered in composting, are known to decrease sorption efficiency [54]. In this study, the biochar from the composts was used at 20 °C, which could benefit from the better sorption of VOCs sorption, observed during the experiment. Although the biochar used is also not typical, as tested before by other authors, it contained only 20% organic matter (Table 1), while others tested >90%. Zhang also emphasised the role of a non-carbonised organic matter content, which in compost biochar could explain the high potential of this biochar for VOC sorption—making up a large share of the mineral fraction, especially for biochars produced in low pyrolysis [54]. Furthermore, they proved that biochars produced at low temperatures (300 °C) have the larger sorption capacity for the adsorption of acetone, cyclohexane, and toluene. Other authors listed the presence of a surface chemical functional group as a source of adsorption control [54]. As identified through ATR-FTIR analysis (Figure 3), the following possible common basic functional groups responsible for VOCs adsorptions were identified in composts’ biochar: carbonyl (C=O, peak 1560); quinones (C-H, peak 1580–1620); pyrrole (N-H:1480-1560, C-N 1190); pyridinium (N-H, peak 1480, 1560); and pyridine-N-oxide (N-O, peak 1000–1300). The possible presence of these groups could also indicate the effectiveness of composts’ biochar. Another possible hypothesis for an explanation of the positive effect of biochar addition is the stimulation of the microorganism metabolism responsible for biodegradation when biochar is added to kitchen waste. Almost 25% of the observed VOC compounds play a role in the metabolism of plants, microorganisms, and humans (e.g., hexanoic acid, methyl ester; benzenemethanol; Appendix A). This could explain the positive effect of biochar on observed VOCs emission.

## 4. Conclusions

The composts’ biochar is a novel form of biochar with a unique composition. Preliminary research has been conducted on the impact of composts’ biochar on volatile organic compound (VOC) emissions during kitchen waste storage, which yielded the following results:The addition of composts’ biochar to kitchen waste demonstrated the significant removal of VOCs, with an over 70% reduction in VOCs emissions when 5–10% biochar was added.The most effective reduction was observed for unpleasant odours such as hexanol <-n> acetic acid, hexyl ester, and diethyl-phthalate, while no effect was observed for pleasant odours such as octanol <n->, limonene, octanoic acid, and benzene-ethanol.The high mineral fraction content of composts’ biochar suggests that the gaseous sorption of VOCs was more controlled by the non-carbonised organic matter content than physical adsorption.In total, 25% of the detected VOC compounds played a role in the plant, microorganism, and human metabolism, suggesting that composts’ biochar has a positive impact on overall metabolism and accelerates biodegradation.

As biochar was successfully used for the elimination of the potential odour components, it should be noted that it could also be dedicated to the sorption of specific dangerous VOCs such as ethylbenzene, o-xylene, acetic acid, and naphthalene, as was observed in other studies with biochar.

Raw biochar contains only a few hazardous compounds, posing a low risk of exposure. Thus, it presents an excellent opportunity for odour elimination in kitchen waste containers. Furthermore, the production of biochar from compost presents a promising opportunity for the development of the local bioeconomy. The production of this type of biochar can help eliminate odours in future processing, such as composting (compost → composts’ biochar → kitchen waste + composts’ biochar → composting of kitchen waste + composts’ biochar).

## Figures and Tables

**Figure 1 materials-16-06413-f001:**
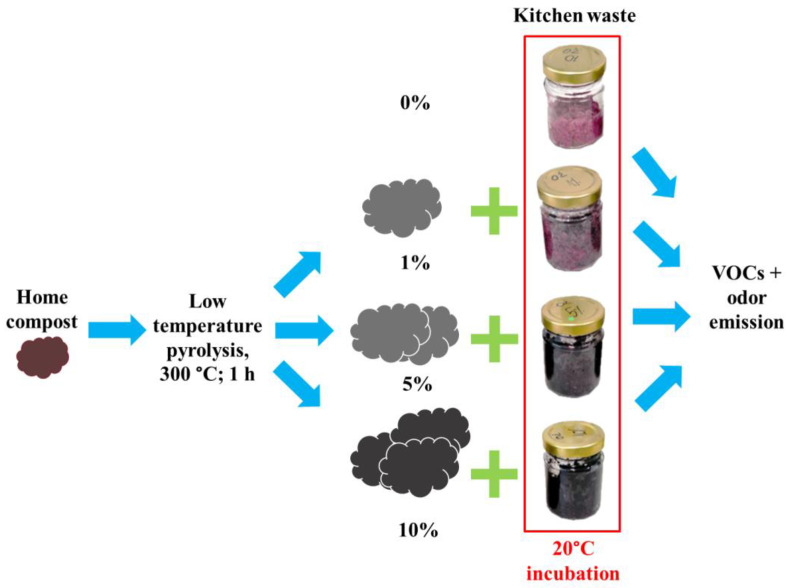
Experiment design.

**Figure 2 materials-16-06413-f002:**
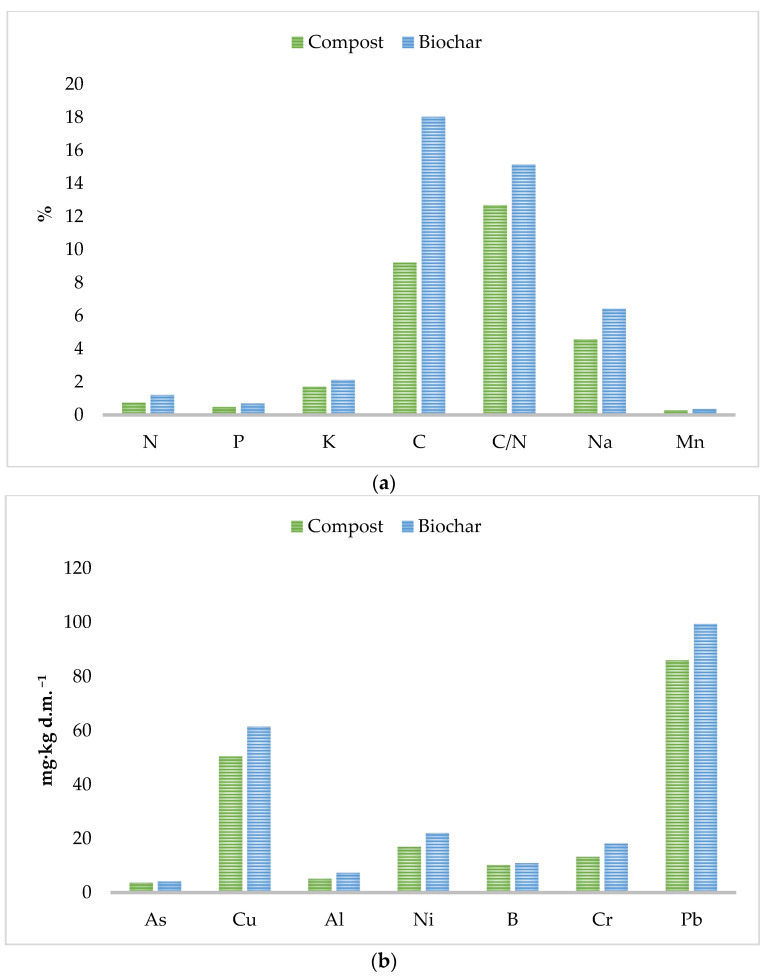
Content of selected elements in compost and biochar used in the experiment: (**a**) essential elements, and (**b**) heavy metals.

**Figure 3 materials-16-06413-f003:**
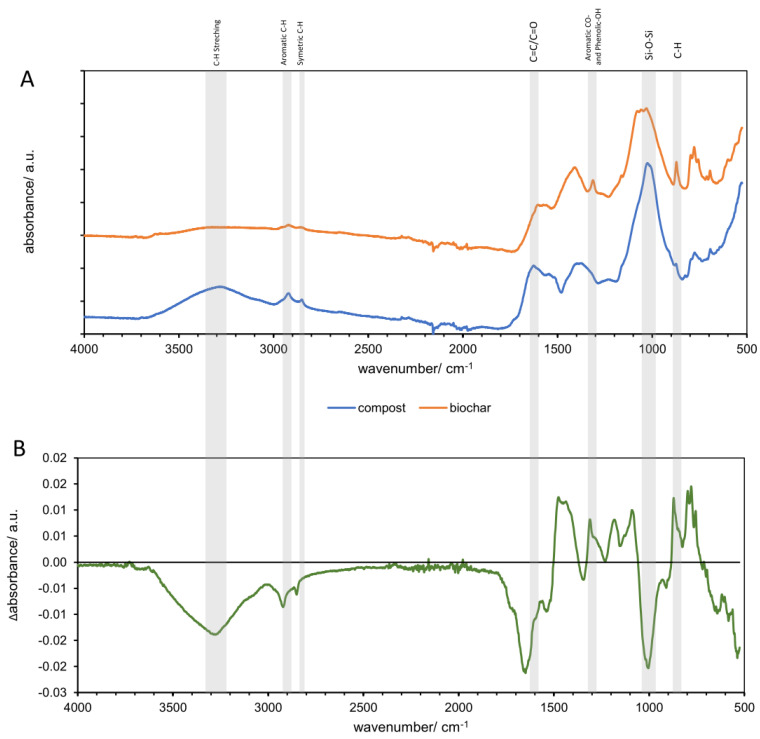
FTIR results of compost and biochar made from compost (**A**) shows the FTIR spectra of compost and biochar made from compost, and (**B**) shows the subtraction spectra of samples. The line indicates zero absorbance change for each subtraction spectrum.

**Figure 4 materials-16-06413-f004:**
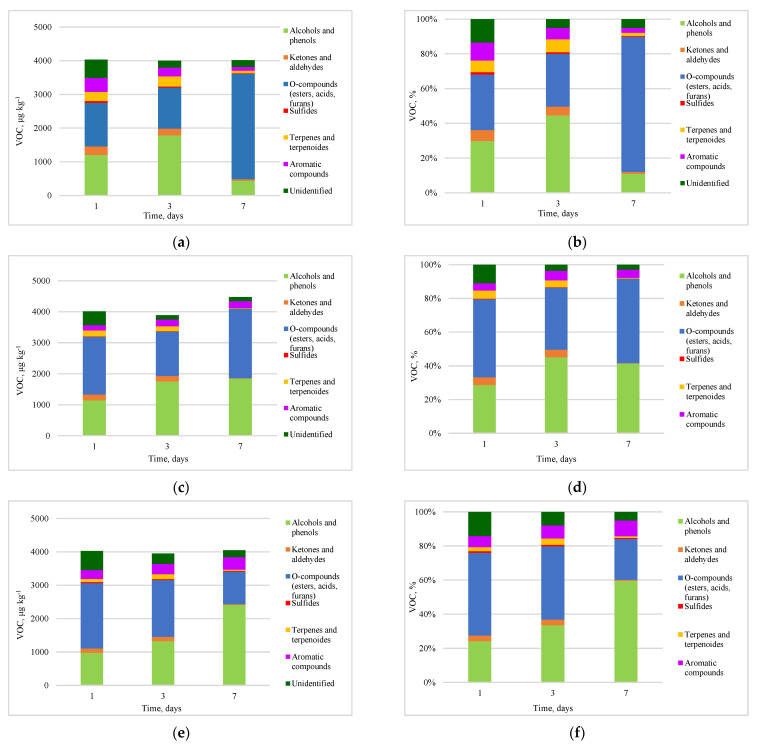
Quantification of the total amount and chemical families of the VOCs identified in kitchen waste during the coffering process (**a**) total emission without the addition of biochar, (**b**) the proportion of each compound category (without biochar), (**c**) total emission with the addition of 1% biochar, (**d**) the proportion of each compound category (1% of biochar), (**e**) total emission with the addition of 5% biochar, (**f**) the proportion of each compound category (5% of biochar), (**g**) total emission with the addition of 10% biochar, (**h**) the proportion of each compound category (10% of biochar).

**Figure 5 materials-16-06413-f005:**
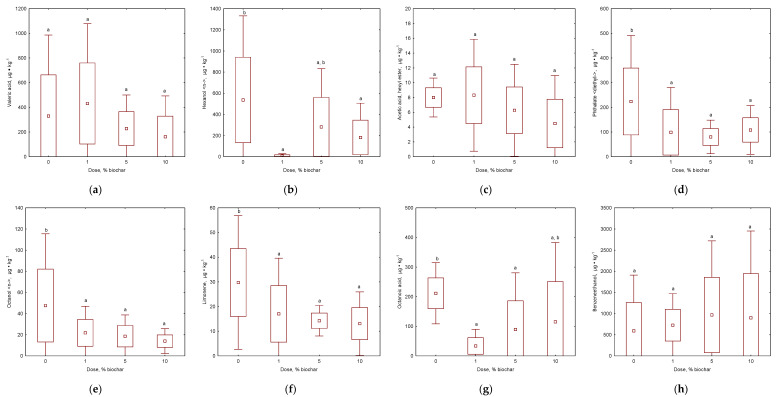
Effect of biochar dose on different VOC emissions: (**a**) Valeric acid; (**b**) Hexanol <-n>; (**c**) Acetic acid, hexyl ester; (**d**) Phthalate <diethyl->; (**e**) Octanol <n->; (**f**) Limonene; (**g**) Octanoic acid; (**h**) Benzene-ethanol. Letters (a, b) indicate the homogeneity group according to Tukey’s test at a significance level *p* < 0.05.

**Figure 6 materials-16-06413-f006:**
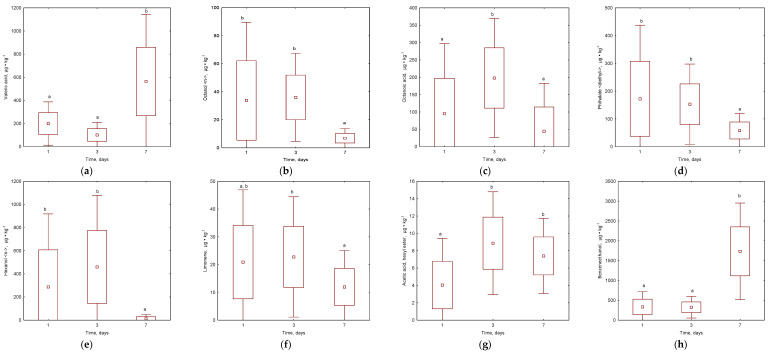
Effect of time on different VOC emissions during the biochar addition: (**a**) Valeric acid; (**b**) Hexanol <-n>; (**c**) Acetic acid, hexyl ester; (**d**) Phthalate <diethyl->; (**e**) Octanol <n->; (**f**) Limonene; (**g**) OctanoCamphoric acid; (**h**) Benzene-ethanol. Letters (a, b) indicate the homogeneity group according to Tukey’s test at significance level *p* < 0.05.

**Table 1 materials-16-06413-t001:** Properties of compost, composts’ biochar, and kitchen waste used in experiment.

Substrate	CEC, cmol(+)‧kg^−1^	N-NH_4_, mg‧kg^−1^ d.m.	N-NO_3_, mg‧kg^−1^ d.m.	LOI, % d.m.	pH, -	Moisture, %	EC, mS‧cm^−1^
Compost	143.28	125.85	1322.03	34	7.09	1.71	2.88
Biochar	123.92	55.99	16.03	19	6.92	40.20	n.d.
Kitchen waste	Day 1				4.92	80.05	2.05
	Day 7				3.87	79.95	1.91

## Data Availability

The data presented in this study are available in Appendix A.

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
