# Peer review of "Application of Composts’ Biochar as Potential Sorbent to Reduce VOCs Emission during Kitchen Waste Storage"

_materials, 2023, doi:10.3390/ma16196413_

Round 1

Reviewer 1 Report

The article deals with testing the effectiveness of compost biochar in order to reduce noxious odours and volatile organic compounds (VOCs) released during kitchen waste storage. The manuscript addresses an issue that is very important and attractive. Results in this manuscript are worthy, but changes should be made.

 The current manuscript needs major revisions.

 Title: Application of Composts’ Biochar as Potential Sorbent for Reduce VOCs Emission during Food Waste Storage

 Article Type: Original paper

 Introduction

Line 50, 57, 64, 71, 82, 97, 212, 221, 241, 250, 317, 337, 359, 381: Please put the reference after mentioning the authors.

Exsemple: „recently reported by Díaz et al. [10]“

Use Reference List and Citations Style Guide for MDPI Journals.

 Materials and methods

Line 130: was the kitchen waste dried before grounding? If it’s not how you did determined the mesh diameter of 3.5 mm?

Line 138: please correct sub and superscripts through whole manuscript.

Line 139: which gas was used to maintain inert atmosphere in the pyrolyitic chamber?

Line 140: which mill was used for grinding?

Line 144: please change “experiments was provided” into “experiments were performed”.

Line 160: CEC methodology is missing.

Line 173: please change “and placed” into “are placed”.

 Since it is hard to follow text, my suggestion is to put abbreviations in the text for compost, biochar, especially samples with biochar concentrations: 0%, 1%, 5% and 10%. (Figure 5 is hard to follow), etc.

 Results and discussion

Line 212: It is not necessary to write all results that Mr Phares obtained in his experiments.

Line 267: please chose and decide was the compost subject to the process of torrefaction or pyrolysis?

Line 275: provide the definition of LOI.

Line 279: “Electrolytic conductivity was examined only in biochar and was 2.88 mScm-1”. In Table 1 is other way around.

Line 307: Table 1 capture is not adequate.

Line 368: please chose and decide is it biocarbon or biochar?

Line 356, 374: why author put “fruit and kitchen waste”, since the fruits are in the kitchen waste?

Line 423, 459: There is no Table S3 in Supplementary materials.

Author Response

Dear reviewer,

Thank you very much for your review. We appreciate your opinion and suggestions. Please find our responses in the attachment. 

Best Regards 

Reviewer 2 Report

I have reviewed the manuscript and identified several areas that require major revisions. Here are my comments with suggested rewrites:

  1. In line 10, it is not clear whether you mean EU or UE regulations. It would be best to write "European Union (EU) or Union Européenne (UE) regulations" to provide both the full form and the abbreviation.
  2. In line 17, please remove the dash and write "and" between 5 and 10%.
  3. Why was the incubation carried out for 7 days? Were there specific reasons for not extending it beyond 7 days? Please provide clarification.
  4. In line 138, please correct "CO2."
  5. There are numerous typos in the materials section. Please carefully proofread and make the necessary corrections.
  6. What kind of feedstock was used for the preparation of biochar (lines 136-139)? Please specify.
  7. What do you mean by "compost-biochar"? Please provide a clear explanation in your methodology section.
  8. Is there a specific reason for using slow-pyrolyzed biochar at 350°C? Please justify this choice. Please rewrite the sentence (lines 164-165) for clarity and accuracy.
  9. Which probability level (5% or 1%) was used to analyze your data (lines 194-199)? Please specify.
  10. The content in Appendix A should be placed in the supplementary materials section.
  11. There are numerous grammatical and structural errors throughout the manuscript. It is strongly recommended to carefully proofread and correct all of them.

Moderate English editings are required.

Author Response

(The authors gave the same response as above.)

Reviewer 3 Report

This work reported application of composts’ biochar as potential sorbent for reduce vocs emission during food waste storage. The study aimed to test the effectiveness of compost biochar in reducing noxious odors and volatile organic compounds (VOCs) released during kitchen waste storage. Varying quantities of compost biochar (0%, 1%, 5%, and 10%) were added to food waste samples and incubated for seven days at 20. VOCs released were analyzed on days 1, 3, and 7 of the storage simulation process. The results indicated that adding 5-10% of composts’ biochar to kitchen waste significantly reduced emissions in 70% of detected VOCs compounds. This study is helpful for waste management, and the manuscript is well prepared. Revisions are suggested to improve the quality of the manuscript.

Grammar errors should be carefully checked and corrected in the manuscript. For example, “Additionally, Compost's biochar” in line 18.

The superscript or subscript should be checked and corrected in the manuscript. For example, line 138: “heating rate of 50°C ∙ min-1. The CO2 was supplied”, line 175 “3 μg of 1 mg‧ml-1 caryophyllene”.

Reference is suggested to support the description of waste management in sustainable way: Conversion of waste plastics into value-added carbon materials. Environmental Chemistry Letters, (2023). 1-32.

The authors should summary the research advance of the field, and point out the research gap. Moreover, the innovation of this work should be clearly described in Introduction.

More details about the sample of kitchen wastes used in this work should be given. The components, the pretreatment, etc.

Figure 1 and figure 2 can be combined, and figure 1 should clearly describe the kitchen wastes (not Material) with the addition of biochar.

The y axis of figure 4 should be checked and corrected.

References are suggested about the background of biochar: Straw-derived biochar for the removal of antibiotics from water: Adsorption and degradation mechanisms, recent advancements and challenges. Environmental Research, (2023). 116998; Phosphorus adsorption by functionalized biochar: A review. Environmental Chemistry Letters, (2023). 21(1), 497-524.

Temperature is an important factor influencing VOCs emmision, and its impact should be studied in this work.

The mechanism of reduce VOCs emission associated with biochar should be revealed in this work.

Grammar errors should be carefully checked and corrected in the manuscript.

Author Response

(The authors gave the same response as above.)

Round 2

Reviewer 1 Report

Thank you, the paper is ready for publication.

Reviewer 2 Report

The authors have successfully incorporated all the comments raised by me. I, therefore accept the paper in present form.

Moderate english editing are require.